# Network Pharmacology and Untargeted Metabolomics Analysis of the Protective Mechanisms of Total Flavonoids from Chuju in Myocardial Ischemia-Reperfusion Injury

**DOI:** 10.3390/ph19010068

**Published:** 2025-12-29

**Authors:** Gaocheng Shi, Huihui Meng, Zongmeng Zhang, Guanglei Zhang, Yanran Li, Hao Yu

**Affiliations:** 1College of Pharmacy, Anhui University of Traditional Chinese Medicine, Hefei 230012, China; sgaocheng@foxmail.com (G.S.); menghuihui0415@foxmail.com (H.M.); liyanran02@foxmail.com (Y.L.); 2College of Traditional Chinese Medicine, Bozhou University, Bozhou 236800, China; zzongmeng@bzuu.edu.cn (Z.Z.); zhguanglei@foxmail.com (G.Z.)

**Keywords:** total flavonoids of Chuju, myocardial ischemia–reperfusion injury, untargeted metabolomics, network pharmacology

## Abstract

**Background/Objectives**: Myocardial ischemia–reperfusion injury (MIRI) is characterized by the exacerbation of tissue damage following the restoration of blood flow to the myocardium. Chuju, recognized for its homology of food and medicine, is derived from the dried capitulum of *Dendranthema morifolium* (Ramat.), cultivated in Chuzhou, Anhui Province, China. Our previous studies have demonstrated that the total flavonoids extracted from Chuju (TFCJ) exhibit pharmacological efficacy against MIRI. This study will further elucidate its protective mechanism. **Methods**: We employed an integrative approach combining untargeted metabolomics, network pharmacology, molecular docking, and in vitro experiments to elucidate the mechanistic basis of TFCJ’s protective effects against MIRI. **Results**: TFCJ protected H9c2 cardiomyocytes from hypoxia/reoxygenation-induced oxidative stress and apoptosis. Integrated analyses identified Nrf2 as a central regulatory node activated by AKT signaling, which, in turn, modulates antioxidant protein expression and glutathione metabolism. Further in vitro experiments demonstrated that TFCJ induced AKT phosphorylation, thereby promoting Nrf2 activation and upregulating HO-1 expression, along with genes involved in glutathione synthesis. **Conclusions**: TFCJ exerts cardioprotective effects by activating the AKT-Nrf2 signaling pathway, regulating the expression of antioxidant and anti-apoptotic genes, and coordinating downstream glutathione metabolism, ultimately maintaining the oxidative-apoptotic balance in myocardial cells.

## 1. Introduction

Myocardial ischemia–reperfusion injury (MIRI) refers to the secondary and complex damage that occurs in myocardial tissue when blood flow is restored following ischemia [1]. Statistical data show that in 2021, 7.25 million people worldwide died from stroke, of which 3.59 million died from ischemic stroke [2]. The pathological process involves multiple factors, including oxidative stress, apoptosis, and inflammatory responses [3], and clinical treatments targeting a single pathway have shown limited efficacy. With its unique multi-component and multi-target synergistic properties, traditional Chinese medicine presents novel strategies for MIRI prevention and treatment [4].

Chuju, derived from the dried capitula of Dendranthema morifolium Ramat., a plant in the Asteraceae family, is a genuine medicinal herb from Chuzhou, Anhui, China [5]. It contains active constituents known as total flavonoids of Chuju (TFCJ), which include quercetin and acacetin [6]. Previous studies have reported its pharmacological benefits, including anti-inflammatory [7], antioxidant [8], and protective effects against cardiac and cerebral ischemic injury [9,10,11]. Our research group’s preliminary studies demonstrated that TFCJ can alleviate myocardial damage in a rat model of MIRI [10]. However, the precise pharmacodynamic basis and molecular mechanisms of TFCJ remain unclear, which significantly limits further progress and clinical use of this medicine.

Elucidating the multi-target mechanisms of traditional Chinese medicine requires modern technologies [12]. Metabolomics can comprehensively capture dynamic changes in endogenous metabolites under pathological conditions or drug intervention, providing direct metabolic phenotype evidence for understanding disease mechanisms [12,13]. Network pharmacology systematically reveals the multidimensional interactions among “drug–component–target–pathway” from a holistic biological network perspective [14,15]. Combining these two approaches aligns with the holistic principles of traditional Chinese medicine and offers a feasible technical strategy to deeply elucidate the mechanisms by which TFCJ combats MIRI.

Accordingly, this study first established an in vitro MIRI model using H9c2 cells that underwent hypoxia/reoxygenation (H/R). By assessing cell viability, oxidative stress markers, and apoptosis-related proteins levels, the cardioprotective effects of TFCJ were verified. Subsequently, an integrated approach combining untargeted metabolomics using UHPLC-QE-MS and network pharmacology was employed to systematically screen for core targets and key metabolic pathways involved. Finally, molecular docking techniques were used to validate interactions between active components and targets. The purpose of this study is to thoroughly elucidate the mechanisms by which TFCJ acts against MIRI, focusing on cellular impacts, metabolic pathways, and molecular networks, thus offering a scientific foundation for its future development and use.

## 2. Results

### 2.1. Preparation and Composition Analysis of TFCJ

Using the mass concentration of rutin (mg/mL) as the horizontal axis and absorbance as the vertical axis, a linear regression equation of the form A = 12.14X + 0.0084 was obtained, with R^2^ = 0.9994. This indicates a good linear relationship between the mass concentration of rutin and absorbance within the measured range. The total flavonoid mass concentration was calculated to be (30.33 ± 3.70) μg/mL. By searching PubChem, reviewing literature reports [16], and analyzing TFCJ monomer data from our previous experiments, we identified a total of 32 compounds with potential cardioprotective effects. Detailed compound information is presented in Table 1 below.

### 2.2. Effect of TFCJ on H/R-Induced Damage in H9c2 Cells

Microscopic observation revealed that H9c2 cells in the normal group had clear contours and were in good overall condition, resembling myoblast-like cells. After modeling, the density of H9c2 cells was significantly reduced, with cells exhibiting shrinkage and widened gaps (Figure 1A). Following pretreatment with TFCJ, the number of H9c2 cells increased markedly compared to the model group, and cell morphology showed significant improvement (Figure 1A). Furthermore, TFCJ pretreatment rescued the reduction in viability of H9c2 cells induced by H/R (Figure 1B). LDH and CK-MB are commonly used biomarkers to assess the extent of cellular damage [17]. In H9c2 cells, H/R induced substantial cellular injury, as evidenced by the elevated levels of LDH and CK-MB (Figure 1C,D). Treatment with TFCJ significantly reduced LDH release and markedly decreased CK-MB levels in H9c2 cells compared to the model group (Figure 1C,D). These results suggest that TFCJ exerts a protective effect on H9c2 cells subjected to H/R injury.

### 2.3. Effects of TFCJ on Oxidative Damage in H9c2 Cells

The level of oxidative stress within cells is typically assessed by measuring changes in oxidative (MDA) and antioxidant (SOD, GSH-Px) markers [18], which reflect the extent of oxidative damage in H9c2 cells. H/R caused a notable rise in MDA levels, along with a significant reduction in SOD and GSH-Px levels (Figure 2A–C). Following treatment with TFCJ, MDA levels were reduced, while SOD and GSH-Px levels increased (Figure 2A–C). Additionally, H/R-induced ROS increase in H9c2 cells was significantly reduced following treatment with various doses of TFCJ (Figure 2D,E). According to qPCR findings, TFCJ treatment markedly boosted the expression levels of SOD1 and GPX4 (Figure 2F). These results indicate that TFCJ effectively reverses oxidative stress imbalance in H9c2 cells undergoing H/R damage, significantly alleviating cellular oxidative damage and demonstrating a clear antioxidant protective effect.

### 2.4. Effect of TFCJ on the Expression of Apoptotic Proteins in H9c2 Cells

H/R treatment significantly impacted the levels of apoptosis-related proteins in H9c2 cells, as demonstrated by Western blot analysis. Specifically, the expression of Bax and cl-caspase-3 proteins was markedly increased, while Bcl-2 expression was significantly decreased (Figure 3A–C). After administering TFCJ, there was a decrease in Bax and cl-caspase-3 protein levels, and an increase in Bcl-2 levels, relative to the model group (Figure 3A–C). qPCR results showed that TFCJ treatment significantly reversed the downregulation of Bcl-2 and Bcl-xL expression induced by H/R (Figure 3D). These findings suggest that TFCJ can regulate the Bcl-2/Bax balance, inhibit the activation of the apoptotic execution protein caspase-3, and that pretreatment with TFCJ significantly suppresses apoptosis in H9c2 cells.

### 2.5. Metabolomic Analysis

#### 2.5.1. PCA and OPLS-DA Analysis

Principal component analysis (PCA) converts the original dataset into a series of orthogonal principal components that collectively characterize the data structure. The outcomes of this analysis are presented in Figure 4A. The first principal component (PCA1) accounts for 49.4% of the total variance, while the second principal component (PCA2) explains 11.3%. Quality control (QC) samples exhibit a tight clustering pattern, demonstrating the reproducibility and stability of the analytical procedure. Furthermore, strong correlations are observed among samples within the same group, with no outliers identified, thereby confirming the suitability of the dataset for subsequent analyses.

Orthogonal partial least squares discriminant analysis (OPLS-DA) was employed to evaluate intra-group clustering and inter-group differentiation of metabolites across the groups, with results presented in Figure 4B. The OPLS-DA score plot revealed a significant separation between the blank control group and the model group, indicating that hypoxia-reoxygenation treatment induced substantial alterations in the metabolic profile of H9c2 cells. Furthermore, effective discrimination was observed between the model group and the TFCJ group, suggesting that TFCJ intervention can ameliorate abnormal cellular metabolic states. Permutation test parameters showed that the comparison between the blank control and model groups yielded R_2_Y = 0.88 and Q_2_ = 0.75; here, R_2_Y reflects the model’s goodness of fit, while Q_2_ represents its predictability. Values closer to 1 indicate superior model performance, and a Q_2_ > 0.5 suggests no overfitting, demonstrating good stability and reliability, thereby accurately characterizing metabolic differences among groups.

#### 2.5.2. Screening of Differential Metabolites

Based on the OPLS-DA model, differential metabolites between groups were screened using the criteria of VIP > 1 and *p* < 0.05, combined with *t*-test *p*-values. Comparing the model group with the control group, and the TFCJ group with the model group, a total of 111 differential metabolites were identified. Among these, 39 metabolites were upregulated, and 72 were downregulated in the model group; administration of TFCJ reversed these metabolite changes. Quantitative values of the differential metabolites were used to calculate a Euclidean distance matrix, followed by clustering using the complete linkage method. The results were visualized as a heatmap, shown in Figure 4C.

#### 2.5.3. Pathway Enrichment Analysis of Differential Metabolites

Differential metabolites were initially mapped to authoritative metabolite databases, including KEGG and PubChem. After obtaining matched information, species-specific pathway databases for *Rattus norvegicus* (rat) were consulted for metabolic pathway analysis. The scoring results of the KEGG pathway enrichment analysis are shown in Table 2. To enable diverse visualization of metabolic changes in the samples, the pathway enrichment bubble plot was transformed into a rectangular tree map, as shown in Figure 4D. Key pathways included pantothenate and CoA biosynthesis, alanine, aspartate, and glutamate metabolism, histidine metabolism, and glycerophospholipid metabolism. These findings suggest that TFCJ may prevent MIRI by modulating these metabolic pathways.

### 2.6. Results of the Network Pharmacology Analysis

#### 2.6.1. Prediction of Active Components and Targets of TFCJ

A total of 18 active components of TFCJ were identified by searching the TCMSP database and reviewing relevant literature (Table 3). Targets of these active components of Chuju total flavonoids were obtained from the PubChem database, yielding a total of 218 targets.

#### 2.6.2. Subsubsection Collection of Targets for the Prevention and Treatment of MIRI by TFCJ

A total of 1929 MIRI-related targets were obtained from databases including OMIM, DrugBank, and GeneCards. The component targets of TFCJ and the disease targets of MIRI were input into an online Venn diagram tool, yielding 107 potential targets for the prevention and treatment of MIRI, as shown in Figure 5A.

#### 2.6.3. Construction of the PPI Network

Potential targets were analyzed for PPI networks using the STRING platform, and the results were displayed using Cytoscape version 3.7.2 (Figure 5B). Based on degree values, 25 core targets were identified, with AKT1, EGFR, PTGS2, GAPDH, and BCL2 exhibiting the highest scores. These targets may play key regulatory roles in the mechanism by which TFCJ counteracts MIRI.

#### 2.6.4. Establishment of the “Drug-Component-Disease-Target” Network

Based on degree values, the ‘Drug-Component-Disease-Target’ network was created (Figure 5C) and analyzed. Eight components, including Boluoside (degree = 59), Acacetin (degree = 57), Naringin (degree = 55), and Quercetin (degree = 54), were identified as potential key active ingredients. AKT1, EGFR, and PTGS2 are likely core targets of TFCJ in combating MIRI. This analysis of the “Drug-Component-Disease-Target” network confirms the multi-component synergistic mechanism of traditional Chinese medicine treatment, highlighting its therapeutic characteristic of acting on multiple targets through multiple active components.

#### 2.6.5. GO and KEGG Enrichment Analysis

GO and KEGG analyses were conducted on the potential target sites. The GO enrichment analysis identified 516 entries, with the top 10 significantly enriched entries ranked and visualized. As shown in Figure 6A, TFCJ may influence MIRI through mechanisms such as negative regulation of the apoptotic process, the ephrin receptor signaling pathway, positive regulation of PI3K/Akt signaling, and response to xenobiotic stimuli. KEGG enrichment analysis identified 172 pathways. The results suggest that TFCJ may mitigate MIRI by modulating pathways including the HIF-1 signaling pathway, endocrine resistance, and EGFR tyrosine kinase inhibitor resistance (Figure 6B).

### 2.7. Results of Molecular Docking

To validate the results of the network pharmacology study, molecular docking was performed using AutoDock Vina 1.1.2 software to analyze the interactions between the screened compounds and key target proteins. The principal active components of TFCJ identified through network pharmacology—quercetin, apigenin, luteolin, acacetin, and naringenin—were docked with core targets, including AKT1, EGFR, PTGS2, BCL2, and GAPDH (Table 4). Docking results exhibiting favorable binding energies were selected for visual presentation (Figure 7). Generally, lower binding energy values indicate stronger receptor-ligand interactions and greater binding stability. The analysis revealed that the core targets AKT1 and EGFR exhibited favorable binding characteristics with the active components.

### 2.8. Integrated Analysis of Metabolomics and Network Pharmacology

The 417 differential metabolites identified through metabolomic screening of H9c2 cells treated with TFCJ for the prevention of MIRI, along with the 107 intersecting targets of TFCJ and MIRI obtained from network pharmacology analysis, were imported into Cytoscape 3.9.1 software using the MetScape plugin. A “target–metabolic pathway–metabolite” network was constructed to integrate metabolomics and network pharmacology data for TFCJ’s prevention of MIRI, as shown in Figure 8. The results indicate that key metabolic pathways—including alanine, aspartate, and glutamate metabolism; glycerophospholipid metabolism; and glutathione metabolism—are closely associated with the 107 critical targets of TFCJ against MIRI. TFCJ might achieve its therapeutic benefits by modulating these key targets, thereby influencing metabolic pathways such as glutathione metabolism and inducing changes in downstream metabolites. The treatment of MIRI by TFCJ might significantly involve these metabolic pathways.

### 2.9. TFCJ Upregulates the Levels of Antioxidant Genes and GSH in H9c2 Cells by Activating AKT-Nrf2

To verify the hypothesis above, we examined changes in the levels of AKT and Nrf2 proteins in H9c2 cells following TFCJ treatment. The results demonstrated that TFCJ significantly induced AKT phosphorylation, indicating activation of the AKT signaling pathway (Figure 9A,B). Additionally, TFCJ markedly increased the protein levels of Nrf2 and its downstream antioxidant protein HO-1 (Figure 9A–C). The data presented in Figure 9D illustrate the changes in glutathione (GSH) content within H/R model cells following TFCJ treatment. The results indicate that induction of the H/R model caused a significant reduction in GSH levels in H9c2 cells, whereas treatment with TFCJ mitigated this decline. Importantly, TFCJ treatment significantly elevated GSH levels in the cytoplasm of H9c2 cells (Figure 9E). qPCR results also showed that *HO-1* mRNA expression levels were restored following TFCJ treatment (Figure 9E). Furthermore, qPCR analysis revealed that the expression of *Slc7a11*, *Gclm*, and *Gclc*—genes involved in GSH synthesis—was significantly upregulated after TFCJ treatment (Figure 9F). In summary, these in vitro results further support our hypothesis that TFCJ exerts cardioprotective effects by activating the AKT-Nrf2 signaling pathway, upregulating antioxidant and anti-apoptotic proteins, and promoting GSH synthesis.

## 3. Discussion

MIRI-induced cardiomyocyte death significantly worsens cardiovascular disease outcomes [1]; restoring the balance between apoptosis and oxidative stress is crucial for halting disease progression [19]. Our previous study has provided preliminary evidence supporting TFCJ’s cardiovascular protective effects [10]. However, the precise molecular mechanisms remain unclear. To understand the protective effects of TFCJ in MIRI, this study integrated network pharmacology, metabolomics, and in vitro validation, taking into account the multi-target and multi-component features of traditional Chinese medicine.

Cellular experiments demonstrated that TFCJ pretreatment increased the survival rate of H9c2 cells following H/R, while reducing the release of myocardial injury-specific markers, LDH, and CK-MB. Additionally, TFCJ markedly upregulated SOD and GSH-Px activities, decreased the production of lipid peroxidation product MDA, and reduced intracellular ROS accumulation. The expression of the anti-apoptotic protein Bcl-2 was increased by TFCJ, whereas the pro-apoptotic proteins Bax and cl-caspase-3 were decreased, as demonstrated by Western blot analysis. Overall, these results indicate that TFCJ confers cardioprotective effects by modulating antioxidant defenses and apoptotic pathways.

Nrf2, as a key antioxidant transcription factor, influences disease progression in MIRI by regulating oxidative stress and ferroptosis [20]. Network pharmacology analyses revealed that TFCJ acts on key targets such as AKT1, EGFR, and BCL2, and is enriched in pathways including the EGFR tyrosine kinase inhibitor resistance and HIF-1 signaling pathway. Notably, these targets and pathways are interconnected within a regulatory network centered on the principal oxidative stress modulator Nrf2. AKT1, a key node in the PI3K/AKT pathway, has been extensively demonstrated to phosphorylate Nrf2 [21], facilitating its release from Keap1 and subsequent nuclear translocation, thereby initiating the transcription of downstream antioxidant genes. Similarly, EGFR signaling can indirectly regulate Nrf2 activity via the PI3K/AKT axis [22]. Nrf2 can also enhance the expression of anti-apoptotic genes in the Bcl-2 family [23]. Furthermore, under conditions of hypoxia and oxidative stress, HIF-1α and Nrf2 engage in mutual regulation; HIF-1α induces Nrf2 expression, while Nrf2 modulates the stability of HIF-1α, collectively influencing glucose metabolism, angiogenesis, and cell survival [24,25]. Molecular docking demonstrated TFCJ’s key active components (such as quercetin, luteolin, and robinin) all demonstrated good binding affinity with potential targets like AKT1, suggesting that these components may participate in activating the Nrf2 signaling pathway. Numerous studies have demonstrated that quercetin, luteolin, and robinin activate the Nrf2 pathway and raise antioxidant enzymes, such as SOD, to scavenge excessive intracellular ROS in cardiomyocytes [26,27,28]. These analyses lead us to propose that TFCJ activates the Nrf2 signaling pathway by modulating AKT1.

Untargeted metabolomics results suggested that metabolic pathways such as glutathione metabolism and glycerophospholipid metabolism are also involved in TFCJ’s anti-MIRI effects. We found that following TFCJ intervention, endogenous metabolites in H9c2 cells changed significantly, with differential metabolites mainly enriched in alanine, aspartate, and glutamate metabolism, glutathione metabolism, glycerophospholipid metabolism, and other pathways. GSH is the most important intracellular antioxidant [29], and Nrf2 is the core transcription factor regulating the rate-limiting enzymes and regeneration-related enzymes of glutathione synthesis [30]. The significant enrichment of glutathione metabolism in differential metabolite analysis directly confirms a strong antioxidant stress response in H9c2 cells under TFCJ intervention, which is highly consistent with activation of the Nrf2 pathway. Additionally, changes in alanine, aspartate, and glutamate metabolism pathways not only provide precursors (such as glutamate) for glutathione synthesis but also reflect reprogramming of nitrogen and energy metabolism [31,32], processes also indirectly regulated by Nrf2. Disruption of glycerophospholipid and sphingolipid metabolism indicates oxidative stress-induced damage to cell membrane structure and potential protective effects of the drug [33,34], while Nrf2 activation has been shown to reduce lipid peroxidation in H9c2 cells treated with TFCJ. Therefore, the “upstream” signal regulation (AKT1-Nrf2) suggested by network pharmacology and the “downstream” metabolic effects (glutathione metabolism, etc.) identified by metabolomics form a perfect complementary relationship, jointly pointing to the Nrf2 pathway as the mechanism of action.

Further in vitro experiments demonstrated that TFCJ induces phosphorylation and activation of AKT, which in turn activates Nrf2, upregulates the expression of HO-1, and promotes the synthesis of GSH. These findings further validate the predictions of network pharmacology and the observed metabolic changes. Upon activation of the Nrf2 pathway, its downstream target gene HO-1 level is typically upregulated [35]. HO-1 produces products with antioxidant, anti-inflammatory, and anti-apoptotic activities by degrading hemoglobin, such as biliverdin/bilirubin and carbon monoxide [36]. qPCR results also showed that TFCJ upregulated the expression levels of genes related to GSH synthesis, regulated by Nrf2. Therefore, we conclude that TFCJ targets and activates the AKT-Nrf2 signaling pathway, thereby promoting the transcription of antioxidant genes like *HO-1*, enhancing glutathione metabolism, scavenging ROS, inhibiting apoptotic signaling, and exerting therapeutic effects.

The cardioprotective effects observed in this study may result from the combined action of multiple active components within TFCJ. However, key questions remain unresolved, including which components play a dominant role and whether synergistic regulatory effects exist among them. Future research will require systematic chemical separation and structural identification techniques (e.g., LC-MS/MS, NMR), combined with pharmacodynamic validation of individual components and their combinations, to thoroughly elucidate the material basis and synergistic mechanisms underlying these effects. Furthermore, the H9c2 cell line, used as an in vitro model, has inherent limitations. Future studies will employ primary cardiomyocytes and in vivo animal models to verify the physiological relevance and generalizability of the observed protective effects. Additionally, we will supplement the in vitro experiments with EC50 data for TFCJ to further refine the completeness and rationality of the drug dose–response relationship. Finally, it should be noted that this study proposed a potential role for the AKT-Nrf2 signaling axis based solely on correlative analysis. Due to the absence of functional validation using AKT inhibitors or Nrf2 gene silencing, a direct causal relationship cannot yet be established. This mechanistic hypothesis awaits confirmation through future interventions such as pharmacological inhibition or gene knockdown/knockout approaches.

## 4. Materials and Methods

### 4.1. Cells and Reagents

H9c2 cardiomyocytes (Cat. No. SCSP-5211) were purchased from the Cell Bank of the Chinese Academy of Sciences (Shanghai, China). Cells were cultured in DMEM culture medium with or without glucose (Cat. No. 11965092, Thermo Fisher Scientific, Waltham, MA, USA), supplemented with 1% penicillin/streptomycin solution (Cat. No. BL505A, Biosharp, Beijing, China) and 10% fetal bovine serum (Cat. No. G247041, Genial Biological Inc., Brighton, CO, USA). Assay kits for lactate dehydrogenase (LDH, Cat. No. 20241116), creatine kinase isoenzymes (CK-MB, Cat. No. H197-1-1), total superoxide dismutase (SOD, Cat. No. 20250320), malondialdehyde (MDA, Cat. No. 20241213), GSH assay kit (Cat. No. A601-1), and glutathione peroxidase (GSH-Px, Cat. No. 20250314) were obtained from Nanjing Jiancheng Bioengineering Institute (Nanjing, China). The CCK-8 kit (Cat. No. C0041) and reactive oxygen species (ROS) detection kit (Cat. No. S0034S) were purchased from Beyotime (Shanghai, China). The antibodies used are as follows: caspase-3 (Cat. No. 24BP241K25, Boster, Wuhan, China), Bcl-2 (Cat. No. 25BP48E07, Boster, Wuhan, China), AKT (Cat. No. 10176-2-AP, Proteintech, Wuhan, China), p-AKT (Cat. No. 80455-1-RR, Proteintech, Wuhan, China), Nrf2 (Cat. No. YT3189, Immunoway Biotechnology, San Jose, CA, USA), Bax (Cat. No. 24XD963H29, Boster, Wuhan, China), and HO-1 (Cat. No. PB0050, Boster, Wuhan, China).

### 4.2. Preparation and Content Determination of TFCJ

Chuju was purchased from Anhui Jutai Chuju Herbal Technology Co., Ltd. Professor Xiangsong Meng, from the College of Traditional Chinese Medicine at Bozhou University, authenticated all samples as genuine. The Chuju was dried and ground into powder. Using the method from Zhao Mengqi et al. [13], the Chuju powder was mixed with 70% ethanol at a material-to-liquid ratio of 1:20 and extracted by ultrasound under the conditions of 60 °C for 40 min. After concentrating by rotary evaporation, D101 macroporous adsorption resin was used to purify the total flavonoids. Impurities were washed away with pure water, and the total flavonoids were subsequently eluted with 70% ethanol and collected.

Rutin reference standard 10.0 mg was dissolved in 70% ethanol and made up to volume in a 50 mL volumetric flask, preparing a rutin reference standard solution with a mass concentration of 0.2 mg/mL. Ten milligrams of freeze-dried TFCJ powder were dissolved in 70% ethanol and made up to volume in a 50 mL volumetric flask to prepare the TFCJ test solution. 0.5, 1.0, 1.5, 2.0, and 2.5 mL of the rutin reference were prepared in 10 mL volumetric flasks. The following solutions (0.4 mL of 5% sodium nitrite, 0.4 mL of 10% aluminum nitrate, and 4 mL of 4% sodium hydroxide) were added sequentially and mixed for the reaction. Finally, the volume was made up to the mark with 70% ethanol and left to stand for 15 min. Using 70% ethanol as the blank, the absorbance was measured at a wavelength of 506 nm. A standard curve of rutin mass concentration (C) versus absorbance (A) was plotted, and the regression equation was calculated. Then, 2.0 mL of the TFCJ test solution was measured at 506 nm following the above procedure. The absorbance value was substituted into the regression equation to calculate the TFCJ content.

### 4.3. Grouping and Model Establishment

H9c2 cells were seeded into culture dishes. Five groups were established: a blank control group (Normal, no treatment), a model group (Model, hypoxia-reoxygenation treatment), a low-dose TFCJ treatment group (TFCJ-L, 0.1 μg/mL + hypoxia-reoxygenation), a medium-dose TFCJ treatment group (TFCJ-M, 1.0 μg/mL + hypoxia-reoxygenation), and a high-dose TFCJ treatment group (TFCJ-H, 10.0 μg/mL + hypoxia-reoxygenation).

The blank control group was cultured under standard conditions at 37 °C in a CO_2_ incubator (BPN-80CH(UV), Yiheng Instruments, Shanghai, China). The other groups received their respective treatments, followed by incubation in glucose-free DMEM medium under hypoxic conditions (37 °C, 1% O_2_, 99% N_2_) for 2 h. Subsequently, the medium was replaced with DMEM containing 10% FBS, and the cells were reoxygenated at conditions (37 °C, 5% CO_2_, 95% air) for 3 h to complete the hypoxia-reoxygenation model.

### 4.4. Cell Viability Assay

A density of 1.8 × 10^4^ cells/mL of H9c2 cells was used in 96-well plates. After the cells had fully adhered, treatments were applied as described above, with the drug groups receiving different doses of TFCJ. Following hypoxia-reoxygenation treatment, 10 μL of CCK-8 working solution was added to each well and incubated at 37 °C in the dark for 60 min. Cell viability was calculated according to the absorbance values measured using a microplate reader.

### 4.5. Measurement of Cellular CK-MB and LDH Levels

A density of 4 × 10^5^ cells/mL of H9c2 cells was used in 6-well plates, with three replicates per group. Grouping and drug administration were performed as described above. After the experiment, culture supernatants were collected by centrifugation at 4000 rpm for 20 min and assayed for CK-MB and LDH levels.

### 4.6. Measurement of Cellular SOD, MDA, GSH-Px, and GSH Levels

A density of 4 × 10^5^ cells/mL of H9c2 cells was used in 6-well plates, with three replicates per group, and treated as described above. After lysing, the lysates were centrifuged at 12,000 rpm for 15 min, and the resulting supernatants were used to measure the levels of SOD, MDA, GSH-*Px*, and GSH.

### 4.7. Detection of ROS

Cells from each treatment group were collected, and the cells were washed with PBS after discarding the culture medium. They were then incubated with 10 μmol/L DCFH-DA in the dark for 20 min. Fluorescence microscopy (Axio Vert A1, Zeiss, Oberkochen, Germany) was used to observe and photograph the cells, with relative fluorescence intensity indicating ROS levels.

### 4.8. qPCR

Total RNA was extracted using TRIzon Reagent (CW0580S, CWBIO, Taizhou, China). RNA concentration was measured with an ultramicro spectrophotometer. The extracted total RNA was reverse transcribed into cDNA and detected using the CFX Connect™ Detection System (BIO-RAD, Hercules, CA, USA). Experimental results were obtained as 2^−ΔΔCT^ values. Primers were synthesized by Sangon Biotech (Shanghai, China). β-actin-F: CCGCGAGTACAACCTTCTTG, β-actin-R: TTCAGGGTCAGGATGCCTC; Slc7a11-F: CGGTGGTGTGTTTGCTGTCTC, Slc7a11-R: CTTGTGGACGTGAATCATGGAGAG; Gclm-F: TGTGTGATGCCACCAGATTTGAC, Gclm-R: GCCTCAGAGAGCAGTTCTTTTGG; Gclc-F: ACCAGTTGGCCACTATCTGC, Gclc-R: GTTCTTCAGGGGCTCCAGTC; SOD1-F: CAAGCGGTGAACCAGTTGTG, SOD1-R: CTCTCTTCATCCGCTGGACC; HO-1-F: GCGAAACAAGCAGAACCCAG, HO-1-R: TACGTAGTGCTGTGTGGCTG; GPX4-F: CTCCATGCACGAATTCGCAG, GPX4-R: GGCATGCAGATCGACTAGCT; Bcl-2-F: CTTCTCTCGTCGCTACCGTC, Bcl-2-R: CAATCCTCCCCCAGTTCACC; Bcl-xL-F: AGGCTGGCGATGAGTTTGAA, Bcl-xL-R: AGAAGAAGGCCACAATGCGA.

### 4.9. Western Blotting

Protein concentration was determined using a BCA assay following cell lysis. The following operating procedures are based on previous experience [37]. Proteins were separated and transferred onto PVDF membranes. After blocking, incubating with primary antibodies (1:1000 dilution) and secondary antibodies (1:2000 dilution), developing, and imaging were performed using a chemiluminescence system, and the results were analyzed with ImageJ software 1.54g.

### 4.10. Metabolomics Analysis

#### 4.10.1. Metabolomics Sample Preparation

Log-phase H9C2 cells were collected, and the culture medium in each well was discarded. Then, 2 mL of pre-cooled PBS buffer was added to each well. After gentle shaking and washing, the PBS was carefully removed. This washing step was repeated twice to thoroughly eliminate residual serum and other interfering components from the culture medium, ensuring the accuracy of subsequent metabolite extraction and detection. After washing, 0.5 mL of 0.25% trypsin-EDTA solution was added to each well, and cells were observed under a microscope in real time. When 70–80% of the cell edges contracted and the cell bodies became rounded, the cells were collected and pipetted. A portion was taken for counting to ensure uniform cell numbers across samples, and the remaining suspension was transferred to a 15 mL enzyme-free centrifuge tube. Centrifugation was performed at 1000 rpm for 5 min at 4 °C. The cells were resuspended in 1 mL of pre-cooled PBS, followed by a second centrifugation under the same conditions for 5 min. The supernatant was discarded again to further purify the cells. The centrifuge tubes were then plunged into liquid nitrogen for 1 min for quenching, after which they were rapidly transferred to a liquid nitrogen tank for storage. Subsequently, the extraction solvent (methanol: acetonitrile: water = 2:2:1, *v*/*v*) was added and thoroughly mixed. After cell disruption for 20 min, samples were centrifuged using the procedure (12,000 rpm, 4 °C, and 15 min). The supernatant was transferred to sample vials for analysis. Equal volumes of supernatants from each sample were mixed to prepare QC samples, which were analyzed alongside the experimental samples to monitor experimental stability.

#### 4.10.2. LC-MS/MS

A Waters ACQUITY UPLC BEH Amide column (2.1 mm × 50 mm, 1.7 μm) facilitated the chromatographic separation. Mobile phase B was acetonitrile. Mobile phase A consisted of water containing 25 mmol/L ammonium acetate and ammonia. The injection volume was 2 μL.

The instrument used was the Vanquish ultra-high-performance liquid chromatograph (Thermo Fisher Scientific). The instrument settings were as follows: auxiliary gas flow rate, 15 Arb; full scan mass spectrometry resolution (Full MS Resolution), 60,000; sheath gas flow rate, 50 Arb; tandem mass spectrometry resolution (MS/MS Resolution), 15,000; capillary temperature, 320 °C; collision energy (Collision Energy), SNCE 20/30/40 (SNCE refers to “Selected Reaction Monitoring-Collision Energy Optimization mode, indicating collision energies set at 20, 30, and 40, respectively); spray voltage (Spray Voltage), 3.8 kV (positive ion mode) or −3.4 kV (negative ion mode).

#### 4.10.3. Data Analysis and Statistical Processing

The raw data files were converted into the mzXML format utilizing ProteoWizard V3.0.24054 software. Metabolite identification was carried out through the application of R packages version 3.5.0 in conjunction with the BiotreeDB database, followed by visualization analyses using R packages. Data were log-transformed and UV-scaled using SIMCA V18.0.1 software. PCA and OPLS-DA modeling were conducted.

### 4.11. Network Pharmacology Analysis

Active components of TFCJ were screened using the Traditional Chinese Medicine Systems Pharmacology Database and Analysis Platform, supplemented by literature. SMILES identifiers were retrieved from the PubChem database, and component targets were obtained via the Swiss Target Prediction database, with target information verified through the UniProt database. Using “myocardial ischemia–reperfusion injury” as the search term, disease targets were collected from DrugBank, OMIM, GeneCards, TTD, and DisGeNET databases. The Venny platform was used to intersect the targets of total flavonoids from Chuju with myocardial ischemia–reperfusion injury disease targets, and a Venn diagram was generated to identify potential targets of total flavonoids against myocardial ischemia–reperfusion injury. The protein–protein interaction (PPI) network was constructed and analyzed using the STRING database and Cytoscape 3.7.2 software. Gene Ontology (GO) function and Kyoto Encyclopedia of Genes and Genomes (KEGG) pathway enrichment analyses of potential targets were performed using the DAVID database, selecting “Official Gene Symbol” and setting species to “Homo sapiens.” The top 10 entries ranked by *p*-value were visualized. Differential metabolites and intersecting targets were input into Cytoscape 3.7.2 software, and the MetScape plugin was used to analyze and visualize the “target-metabolic pathway-metabolite” network.

### 4.12. Molecular Docking

The SDF format structures of TFCJ were obtained from the PubChem database, and the PDB format files of target proteins were retrieved from the RCSB PDB database, including AKT1 (PDB ID: 7APJ), EGFR (PDB ID: 3NJP), PTGS2 (PDB ID: 5IKR), BCL2 (PDB ID: 6O0K), and GAPDH (PDB ID: 8WWZ). Using AutoDock Tools 1.5.6 software, the ligand and protein files were preprocessed by removing water molecules and adding polar hydrogens, then saved in PDBQT format. Grid boxes were defined to analyze the binding sites of each protein. For the AKT1 protein, a grid box measuring 56 × 74 × 104 Å was constructed, centered at coordinates (X: 3.322, Y: 8.266, Z: −5.171 Å); for the EGFR protein, a grid box of 108 × 112 × 72 Å was centered at (X: 86.765, Y: 48.609, Z: 62.448 Å); for the PTGS2 protein, a grid box of 66 × 82 × 76 Å was centered at (X: 33.269, Y: 23.356, Z: 70.092 Å), ensuring full coverage of the binding pocket. Molecular docking was performed using AutoDock Vina 1.1.2, and structural visualization was conducted with PyMOL 2.2.0 software.

### 4.13. Statistical Analysis

Statistical analysis and graphing were performed using GraphPad Prism version 8. Data are presented as mean ± standard deviation. One-way analysis of variance (ANOVA) was applied to data that met the assumptions of normality and homogeneity of variance. A *p*-value < 0.05 was considered statistically significant.

## 5. Conclusions

In conclusion, this study utilized network pharmacology, untargeted metabolomics, and molecular docking techniques to clarify that TFCJ, through active components such as quercetin, regulates 111 differential metabolites and 25 core targets in H9c2 cells, synergistically modulating the AKT-Nrf2 signaling pathway. This alleviates oxidative stress, inhibits cell apoptosis, and ameliorates metabolic disorders to exert anti-MIRI effects. This research demonstrates the therapeutic benefits of the ‘multi-component, multi-target’ approach in traditional Chinese medicine but also provides a clear molecular regulatory network for the mechanism of TFCJ against MIRI, laying a foundation for subsequent studies.

## Figures and Tables

**Figure 1 pharmaceuticals-19-00068-f001:**
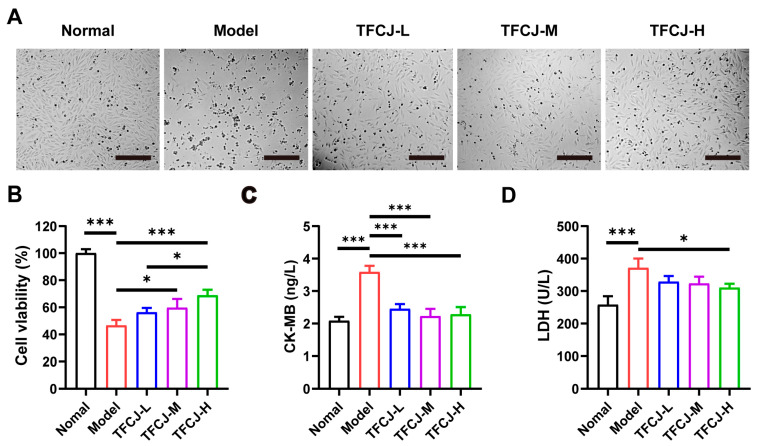
TFCJ alleviates H/R-induced damage in H9c2 cells. (**A**) Representative images of H9c2 cells observed under bright-field microscopy. Scale bar: 100 μm. (**B**) Effect of TFCJ on the viability of H9c2 cells (*n* = 3). (**C**,**D**) Effect of TFCJ on CK-MB and LDH levels in H9c2 cells (*n* = 3). All experiments were performed in triplicate. * *p* < 0.05, *** *p* < 0.001, one-way ANOVA.

**Figure 2 pharmaceuticals-19-00068-f002:**
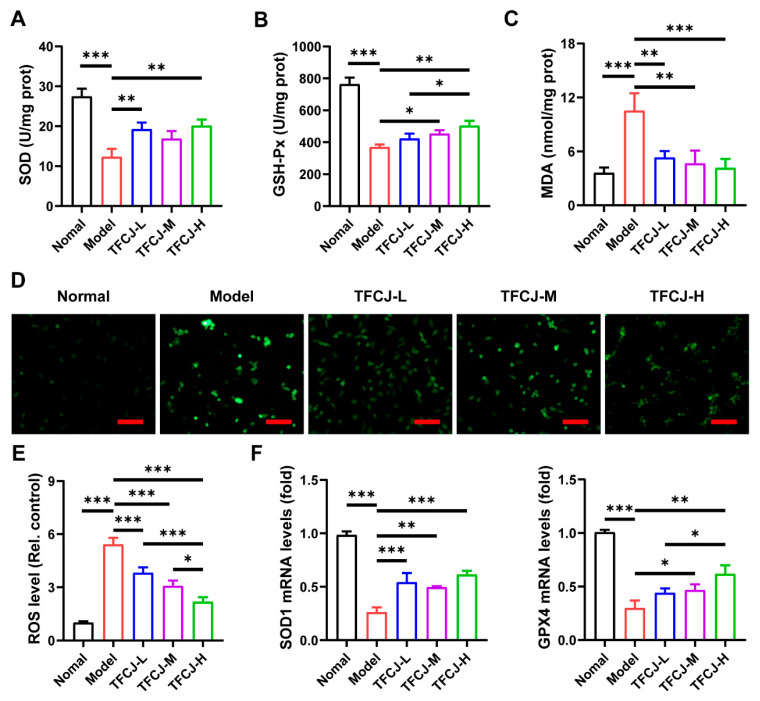
Effect of TFCJ on ROS levels in H9c2 Cells subjected to H/R injury. Effect of TFCJ on (**A**) SOD, (**B**) MDA, and (**C**) GSH-Px levels in H9c2 cells (*n* = 3). (**D**,**E**) Intracellular ROS were labeled and quantified using DCFH-DA (*n* = 3). Scale bar: 50 μm. (**F**) qPCR analysis of the expression levels of SOD1 and GPX4 (*n* = 3). All experiments were performed in triplicate. * *p* < 0.05, ** *p* < 0.01, *** *p* < 0.001, one-way ANOVA.

**Figure 3 pharmaceuticals-19-00068-f003:**
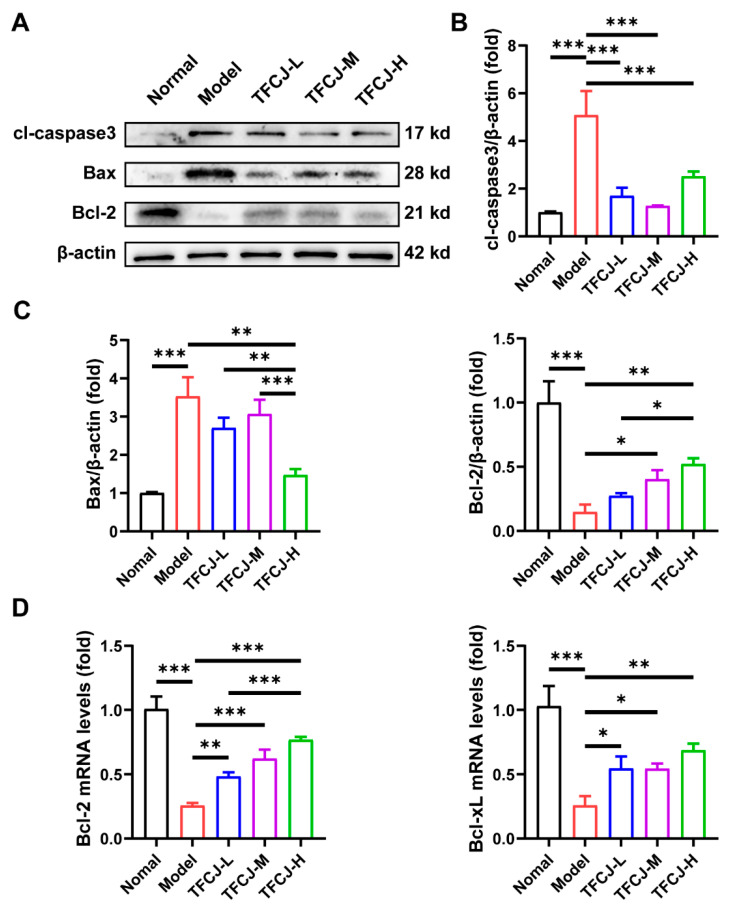
Expression levels of apoptosis-related proteins in H9c2 cells. (**A**–**C**) Western blot analysis of cl-caspase3, Bcl-2, and Bax protein levels in H9c2 cells (*n* = 3). (**D**) qPCR analysis of the expression levels of Bcl-2 and Bcl-xL (*n* = 3). All experiments were performed in triplicate. * *p* < 0.05, ** *p* < 0.01, *** *p* < 0.001, one-way ANOVA.

**Figure 4 pharmaceuticals-19-00068-f004:**
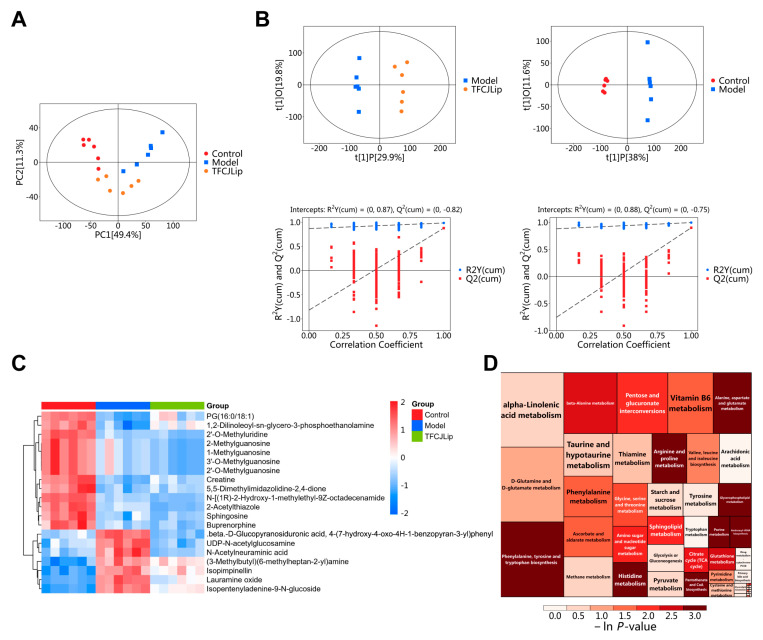
(**A**) PCA. (**B**) OPLS and permutation test plots for each group. (**C**) Cluster heatmap of differential metabolites. (**D**) Metabolic pathways related to TFCJ’s protection against MIRI.

**Figure 5 pharmaceuticals-19-00068-f005:**
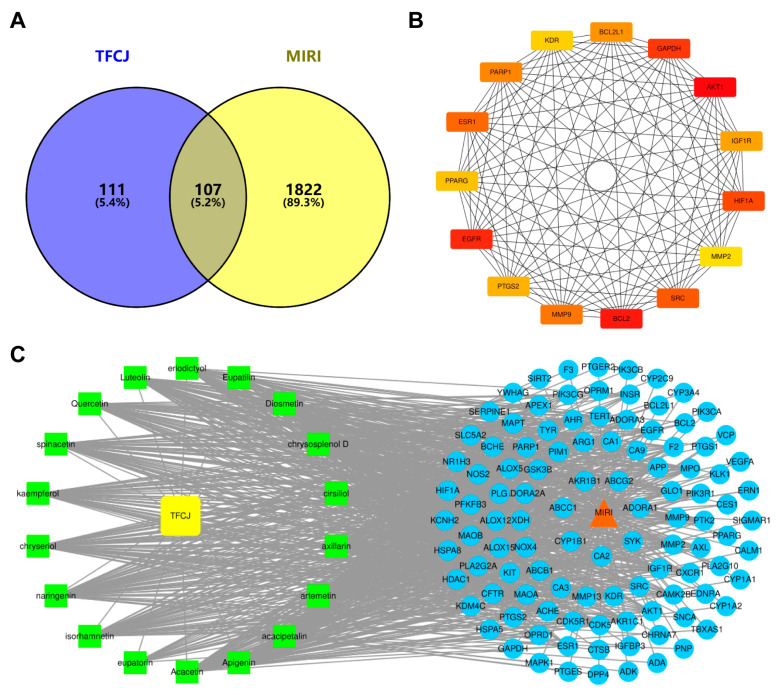
(**A**) Venn diagram of total flavonoids from Chuju in the prevention and treatment of MIRI. (**B**) PPI network diagram. Note: In (**B**), the darker the node color, the higher the degree value; more connections indicate a stronger association with the disease. (**C**) “Drug-Component-Disease-Target” network diagram, yellow and green squares represent TFCJ and active components, respectively; red arrow shapes denote the disease MIRI; blue circles indicate targets.

**Figure 6 pharmaceuticals-19-00068-f006:**
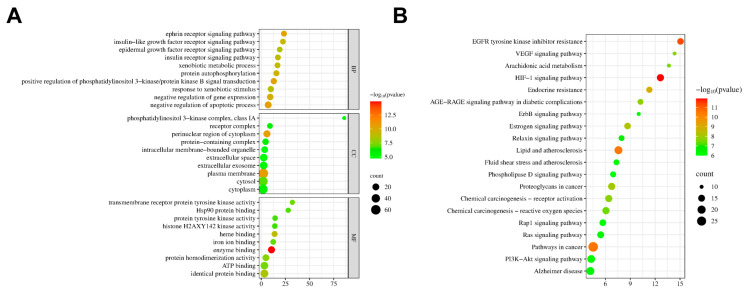
(**A**) GO enrichment analysis of TFCJ against MIRI. (**B**) KEGG pathway enrichment analysis.

**Figure 7 pharmaceuticals-19-00068-f007:**
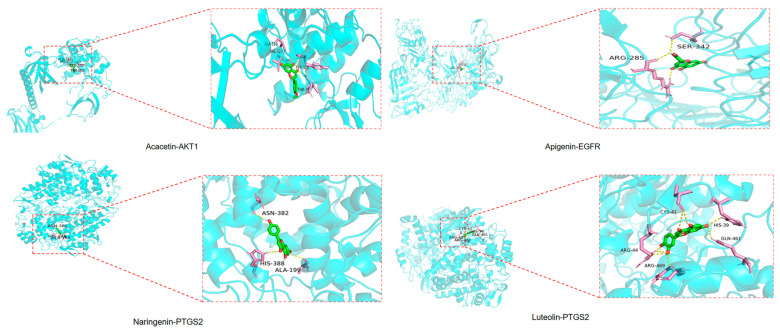
Molecular docking pattern diagram of TFCJ and key targets.

**Figure 8 pharmaceuticals-19-00068-f008:**
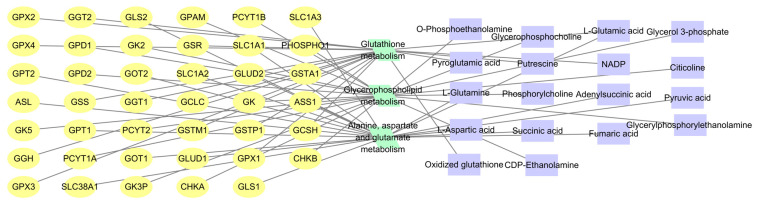
“Target-Metabolite-Metabolic Pathway” Network.

**Figure 9 pharmaceuticals-19-00068-f009:**
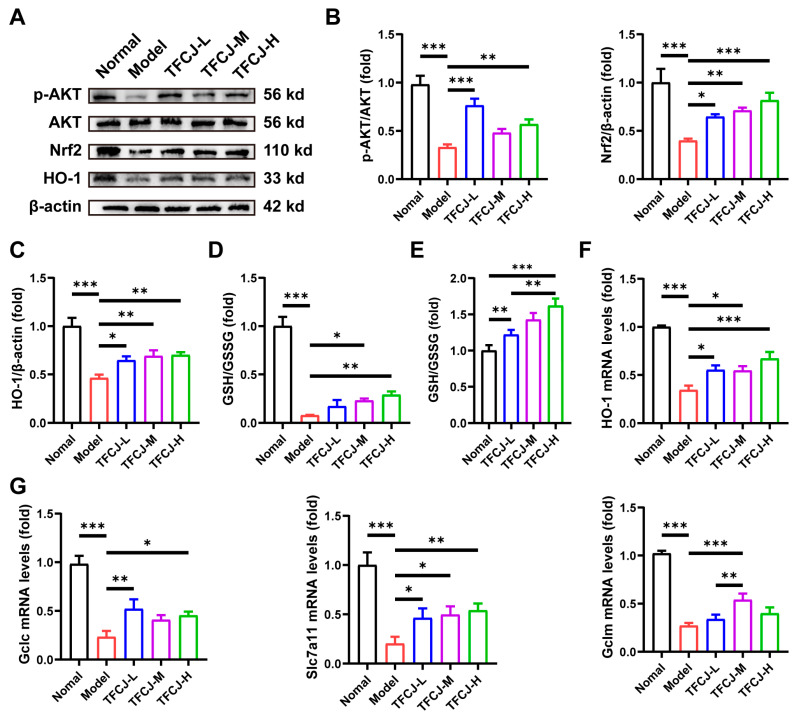
TFCJ upregulates the levels of antioxidant genes and GSH in H9c2 cells by activating AKT-Nrf2. (**A**–**C**) Western blot and quantification analysis of p-AKT, Nrf2, and HO-1 protein levels (*n* = 3). (**D**) Analysis of GSH content in H9c2 cells subjected to hypoxia/reoxygenation (H/R) after treatment with TFCJ (*n* = 3). (**E**) Analysis of GSH content in H9c2 cells following TFCJ treatment. qPCR analysis of the expression levels of (**F**) *HO-1*, (**G**) *Gclc*, *Gclm*, and *Slc7a11* (*n* = 3). * *p* < 0.05, ** *p* < 0.01, *** *p* < 0.001, one-way ANOVA.

**Table 1 pharmaceuticals-19-00068-t001:** The main flavonoid components of Chuju.

No.	Compounds	Formula	mz
1	Eriodictyol 7-rhamnoside	C_21_H_22_O_10_	433.11
2	Luteolin 6-C-glucoside 8-C-arabinoside	C_27_H_30_O_16_	609.16
3	Quercetin 3-O-malonylglucoside	C_24_H_22_O_15_	551.1
4	Kaempferol 3-glucuronide	C_21_H_18_O_12_	461.08
5	Rutin	C_27_H_30_O_16_	609.16
6	quercetin 3-O-glucuronide	C_21_H_18_O_13_	479.08
7	Eriodictyol	C_15_H_12_O_6_	289.07
8	quercetin	C_15_H_10_O_7_	303.05
9	Cyanidin-3-O-glucoside	C_21_H_21_O_11_	450.11
10	Luteolin 7-glucoside	C_21_H_20_O_11_	447.1
11	Quercetin 3-(6′′-malonyl-glucoside)	C_24_H_22_O_15_	551.1
12	Diosmetin-7-O-rutinoside	C_28_H_32_O_15_	609.17
13	Diosmetin	C_16_H_12_O_6_	301.07
14	Isorhamnetin	C_16_H_12_O_7_	317.07
15	Apigenin	C_15_H_10_O_5_	269.05
16	Linarin	C_28_H_32_O_14_	593.18
17	Luteolin	C_15_H_10_O_6_	287.05
18	Acacetin	C_16_H_12_O_5_	285.07
19	Wogonoside	C_22_H_20_O_11_	461.11
20	D-Phenylalanine	C_9_H_11_NO_2_	166.08504
21	Tricin	C_17_H_14_O_7_	331.08014
22	Pachypodol	C_18_H_16_O_8_	361.0894
23	Nerolidol	C_15_H_26_O	205.19019
24	Artemitin	C_20_H_20_O_8_	389.1181
25	Parthenolide	C_15_H_20_O_3_	231.13742
26	Phthalic acid	C_8_H_6_O_4_	149.02257
27	Quinic acid	C_7_H_12_O_6_	191.05876
28	5-Caffeoylquinic acid	C_16_H_18_O_9_	353.09726
29	Spiraeoside	C_21_H_20_O_12_	301.03964
30	5,7,3′,4′-Tetrahydroxy-6,8-dimethoxyflavone	C_17_H_14_O_8_	345.06622
31	Idoxanthin	C_40_H_54_O_4_	597.37714
32	Skullcapflavone II	C_19_H_18_O_8_	373.0985

**Table 2 pharmaceuticals-19-00068-t002:** Top 10 KEGG pathway analysis results.

Pathway Name	*p*	−log (*p*)	FDR	Impact
Pantothenate and CoA biosynthesis	0.0018256	6.3059	0.14787	0.12245
Phenylalanine, tyrosine, and tryptophan biosynthesis	0.0080153	4.8264	0.23773	1
Amino sugar and nucleotide sugar metabolism	0.011065	4.5039	0.23773	0.18802
Pentose and glucuronate interconversions	0.013826	4.2812	0.23773	0.63637
Glutathione metabolism	0.014675	4.2216	0.23773	0.04007
Glycerophospholipid metabolism	0.024055	3.7274	0.32475	0.17407
Ascorbate and aldarate metabolism	0.042584	3.1563	0.43117	0.4
Phenylalanine metabolism	0.042584	3.1563	0.43117	0.40741
Ubiquinone and other terpenoid-quinone biosynthesis	0.10925	2.2141	0.98326	0
beta-Alanine metabolism	0.15942	1.8362	1	0.22222

**Table 3 pharmaceuticals-19-00068-t003:** The screened flavonoids of Chuju.

Compound	Molecule Formula	Compound	Molecule Formula
Acacetin	C_16_H_12_O_5_	Eriodictyol	C_15_H_12_O_6_
Apigenin	C_15_H_10_O_5_	Luteolin	C_15_H_10_O_6_
Acacipetalin	C_11_H_17_NO_6_	Quercetin	C_15_H_10_O_7_
Artemetin	C_20_H_20_O_8_	Spinacetin	C_17_H_14_O_8_
Axillarin	C_17_H_14_O_8_	Kaempferol	C_15_H_10_O_6_
Cirsiliol	C_17_H_14_O_7_	Chryseriol	C_16_H_12_O_6_
Chrysosplenol D	C_18_H_16_O_8_	Naringenin	C_15_H_12_O_5_
Diosmetin	C_16_H_12_O_6_	Isorhamnetin	C_16_H_12_O_7_
Eupatilin	C_18_H_16_O_7_	Eupatorin	C_18_H_16_O_7_

**Table 4 pharmaceuticals-19-00068-t004:** Molecular binding energy between TFCJ and key targets.

Compound	Gene	Binding Energy/kcal/mol	Compound	Gene	Binding Energy/kcal/mol
Acacetin	PTGS2	−9.30	Luteolin	PTGS2	−9.50
	AKT1	−9.40		AKT1	−7.20
	EGFR	−9.10		EGFR	−8.90
	GAPDH	−8.40		GAPDH	−8.90
	BCL2	−7.20		BCL2	−7.30
Naringenin	PTGS2	−9.30	Apigenin	PTGS2	−9.00
	AKT1	−7.10		AKT1	−7.20
	EGFR	−8.90		EGFR	−8.90
	GAPDH	−9.00		GAPDH	−9.00
	BCL2	−7.00		BCL2	−7.30

## Data Availability

The original contributions presented in this study are included in the article. Further inquiries can be directed to the corresponding author.

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
