# Peer review of "Network Pharmacology and Untargeted Metabolomics Analysis of the Protective Mechanisms of Total Flavonoids from Chuju in Myocardial Ischemia-Reperfusion Injury"

_pharmaceuticals, 2025, doi:10.3390/ph19010068_

Round 1

Reviewer 1 Report

Comments and Suggestions for Authors

The authours discussed the topic of "Network Pharmacology and Untargeted Metabolomics Analysis of the Protective Mechanisms of Total Flavonoids from Chuju in Myocardial Ischemia-Reperfusion Injury". I have some quesions for consideration:

(1) Why this study use Chuju? What is the significant, although it is a homology of food and medicine.

(2) Why use rutin as a standard reference in Chuju? Is this belong to the Chinese Pharmacopoeia?

(3) There are 32 compounds in Chuju, it only focus on the rutin? Is this the only bioactive compound in Chuju? Another such as quercetin, luteolin, and apigenin.

(4) What is the best concentration or gram of Chuju used in Myocardial ischemia reperfusion injury?

(5) Please discuss detail for the mechanism of TFCJ on Oxidative Damage in H9c2 Cells. 

(6) What is the important of using Orthogonal partial least squares discriminant analysis? Are there any relationship between "Screening of Differential Metabolites" and "Pathway Enrichment Analysis of Differential Metabolites"? 

(7) Please check the format in manuscript. Generally, it should have write experimental produces first, then result, discussion, conclusion. 

Author Response

The authours discussed the topic of "Network Pharmacology and Untargeted Metabolomics Analysis of the Protective Mechanisms of Total Flavonoids from Chuju in Myocardial Ischemia-Reperfusion Injury". I have some quesions for consideration:

Comment 1:Why this study use Chuju? What is the significant, although it is a homology of food and medicine.

Response 1: We appreciate your insightful inquiry concerning the selection criteria of Chuju in this investigation. The decision to focus on Chuju was deliberate and grounded in preliminary experimental validation, the identification of a distinct research gap, and its recognized dual functionality as both a food and medicinal resource. Our research group has maintained a sustained interest in the cardioprotective properties of medicinal herbs exhibiting this duality. Prior animal studies employing the myocardial ischemia-reperfusion injury (MIRI) model in Sprague-Dawley rats have systematically demonstrated the biological activity of Chuju [1]. Nonetheless, the precise mechanisms underlying these effects have yet to be elucidated. Accordingly, the principal aim of the present study is to address this gap by comprehensively characterizing the molecular mechanisms through which Chuju exerts its anti-MIRI effects, utilizing approaches such as network pharmacology, molecular docking, and experimental validation. This endeavor seeks to establish a robust scientific foundation for its clinical application. Furthermore, compared to conventional chemical pharmaceuticals, medicinal herbs with combined food and medicinal properties present enhanced safety profiles, rendering them particularly suitable for the prevention and adjunctive treatment of myocardial ischemia-reperfusion injury, thereby possessing considerable translational potential. Thus, the selection of Chuju as the focal subject of this research is congruent with the objective of elucidating the anti-MIRI mechanisms of its total flavonoids and contributes to the advancement of novel, safe, and efficacious cardioprotective therapeutics.

Comment 2:Why use rutin as a standard reference in Chuju? Is this belong to the Chinese Pharmacopoeia?

Response 2: This study employed rutin as the standard reference substance, aligning with the regulatory criteria established in the first edition of the Chinese Pharmacopoeia and adhering to prevalent research conventions within the field [2]. As a prototypical natural flavonoid compound, rutin offers several advantages, including structural stability, straightforward purity control, and characteristic ultraviolet absorption properties. Consequently, it has been widely adopted as the universal standard reference for quantifying total flavonoid content in plant materials across both domestic and international studies. The selection of rutin as the reference standard thus ensures the precision, regulatory compliance, and comparability of the results obtained in total flavonoid content determination.

Comment 3: There are 32 compounds in Chuju, it only focus on the rutin? Is this the only bioactive compound in Chuju? Another such as quercetin, luteolin, and apigenin.

Response 3: Thank you for your inquiry. Our study identifies quercetin, luteolin, and apigenin as the primary active functional constituents of TFCJ. Rutin is employed exclusively as a reference compound for the construction of the standard curve utilized in quantifying the content of the extracted TFCJ.

Comment 4:What is the best concentration or gram of Chuju used in Myocardial ischemia reperfusion injury?

Response 4: In our previous study, the optimal therapeutic concentration of TFCJ in the MIRI rat model was 40 mg/kg [1]. In the present study, we found that a low dose of 0.1 μg/ml could exert a cardioprotective effect in vitro. However, the effects of low, medium, and high doses on reducing myocardial injury were inconsistent across various indicators. Further optimization of the TFCJ dosing regimen may be required to establish the dose-response relationship and determine the optimal administration dose.

Comment 5: Please discuss detail for the mechanism of TFCJ on Oxidative Damage in H9c2 Cells.

Response 5: Thank you for your suggestion. The discussion section of our manuscript already includes the mechanism by which TFCJ potentially alleviates myocardial oxidative damage induced by MIRI through activation of the AKT-Nrf2 signaling pathway (see lines 288 to 316 on page 13 of the manuscript).

Comment 6: What is the important of using Orthogonal partial least squares discriminant analysis? Are there any relationship between "Screening of Differential Metabolites" and "Pathway Enrichment Analysis of Differential Metabolites"?

Response 6: This study employed orthogonal partial least squares discriminant analysis (OPLS-DA) for the analysis of metabolomics data due to its capability to quantify the contribution of individual metabolites to group differentiation through the calculation of variable importance in projection (VIP) scores. When combined with univariate statistical methods, this approach facilitates the rapid identification of key metabolites that genuinely drive group differences from a large metabolite dataset, thereby mitigating the susceptibility to interference and reducing the high false-positive rates commonly associated with traditional analytical techniques. This methodology establishes a robust data foundation for subsequent pathway enrichment analysis and mechanistic exploration.

Pathway enrichment analysis serves as a mechanistic deepening step that relies directly on the accurately screened differential metabolites; these two processes are intrinsically linked and build progressively upon one another. Without the precise prior identification of differential metabolites, pathway enrichment analysis would produce inconsequential results due to the overwhelming number of metabolites involved. Conversely, in the absence of pathway enrichment analysis, the differential metabolites identified cannot independently elucidate the metabolic alterations occurring during the pathological progression of MIRI or the modulatory effects of TFCJ on H9c2 cardiomyocytes.

Comment 7:Please check the format in manuscript. Generally, it should have write experimental produces first, then result, discussion, conclusion.

Response 7: Thank you for your suggestion. We have organized the manuscript in accordance with the journal's submission template.

Reference

  1. Hao, Y.; Xin, X.; Han-zhen, L.; Jin-xing, Z.; Shi-tang, M.; Xiao-lin, Z. Effect of Total Flavonoids from Dendranthema morifolium (Ramat) Tzvel. cv. Chuju Flowers on Myocardial Ischemia Reperfusion Injury in Rats. Food Sci. 2012, 33, 283-286.
  2. Chinese Pharmacopoeia Commission. Pharmacopoeia of the People’s Republic of China [ChP]. Vol. I. Beijing: China Medical Science Press; 2020.

Reviewer 2 Report

Comments and Suggestions for Authors

The manuscript presents a multi-omics investigation (untargeted metabolomics, network pharmacology, molecular docking) combined with in-vitro experiments to elucidate how total flavonoids from Chuju (TFCJ) protects H9c2 cells from hypoxia/reoxygenation (H/R) injury. The work is valuable and aligns with current trends in Traditional Chinese Medicine mechanistic studies. However, several aspects need improvement mainly in methodological rigor, data interpretation, statistical reporting, and clarity of writing. To the best of my knowledge, I summarize below points for the improvement of the manuscript-

  • The combination of network pharmacology, metabolomics, molecular docking, and in-vitro validation strengthens the mechanistic claims. The integration figure (Figure 8) nicely visualizes pathway-target-metabolite relationships. 111 differential metabolites and several metabolic pathways were identified (Figure 4), which provides novel biochemical insights into TFCJ's impact on MIRI models.
  • Although Table 1 lists 32 flavonoids, the study does not quantify the proportion of each component neiher ensure batch consistency or standardization. Also there are no chromatograms for quality control. Please provide LC-MS chromatograms, quantitative profiles, and QC metrics. Discuss batch reproducibility since herbal extracts vary significantly.
  • H9c2 is not a true cardiomyocyte and lacks several cardiac metabolic features. The model is widely used but should be contextualized. Add a paragraph in the Discussion acknowledging limitations of H9c2 cells and suggesting future validation in primary cardiomyocytes or in vivo models.
  • Metabolomics Statistical Rigor Needs Improvement. No mention of correcting p-values for multiple testing (FDR). VIP>1 and p<0.05 without FDR inflates false positives. PCA plots are missing, only OPLS-DA shown. Permutation test outputs need clearer presentation.
  • I would suggest to include unsupervised PCA plots. Add QC sample clustering results to demonstrate analytic stability.
  • Network Pharmacology Target Prediction Is Too Broad. The manuscript identifies 1,929 MIRI-related targets and 218 compound targets, resulting in 107 overlapping targets. These numbers are excessively high and likely include false positives.
  • Use more strict filtering criteria (e.g., probability >0.7 in SwissTargetPrediction). Justify why so many predicted targets are biologically relevant. Perform topological importance analysis with more stringent thresholds.
  • Molecular Docking is not sufficiently robust. Docking results are shown only as binding energy values. The some important elements are missing like- RMSD validation; description of active site preparation; positive control ligand docking; Binding energies across different proteins are not directly comparable.
  • I would suggest to add docking controls and provide docking poses with key interactions labeled. Report grid box parameters.
  • In-Vitro experiments lack dose–response justification. The doses chosen (0.1, 1.0, 10 μg/mL) lack justification (are they physiologically relevant?). Also authors do not explain why non-linear response trends have been chosen. Discuss dose selection rationale and consider including EC50 data.
  • The authors conclude that AKT activation is the primary upstream regulator of Nrf2 activation; however No AKT inhibitor or Nrf2 siRNA was used to validate causality. Thus, mechanistic claims remain correlational. Clearly state the limitation and avoid implying proven causality.
  • Increase resolution for Figures 1–3. Figure legends should also specify statistical tests used.
  • Also add details on: Number of biological vs. technical replicates; whether experiments were blinded or Randomization procedures (if any).
  • Provide KEGG pathway impact scores. Clarify how metabolites were identified (MS/MS match score, library used).
  • Current metabolomics is untargeted. My suggestion would be that targeted validation will improve credibility. If possible, include that.
  • Page 2, line 54; cite the latest reference in support of your statement https://doi.org/10.1016/j.ymgme.2025.109180

Author Response

Reviewer #2:

The manuscript presents a multi-omics investigation (untargeted metabolomics, network pharmacology, molecular docking) combined with in-vitro experiments to elucidate how total flavonoids from Chuju (TFCJ) protects H9c2 cells from hypoxia/reoxygenation (H/R) injury. The work is valuable and aligns with current trends in Traditional Chinese Medicine mechanistic studies. However, several aspects need improvement mainly in methodological rigor, data interpretation, statistical reporting, and clarity of writing. To the best of my knowledge, I summarize below points for the improvement of the manuscript.

The combination of network pharmacology, metabolomics, molecular docking, and in-vitro validation strengthens the mechanistic claims. The integration figure (Figure 8) nicely visualizes pathway-target-metabolite relationships. 111 differential metabolites and several metabolic pathways were identified (Figure 4), which provides novel biochemical insights into TFCJ's impact on MIRI models.

Comment 1: Although Table 1 lists 32 flavonoids, the study does not quantify the proportion of each component neiher ensure batch consistency or standardization. Also, there are no chromatograms for quality control. Please provide LC-MS chromatograms, quantitative profiles, and QC metrics. Discuss batch reproducibility since herbal extracts vary significantly.

Response 1: Thank you for your reminder and suggestions. We sincerely apologize for the misunderstanding. Since the experiments, data compilation, and manuscript writing were conducted by different students, we mistakenly believed that the 32 flavonoids listed were identified through LC-MS analysis. In fact, the 32 flavonoids we listed are based on our previous research results and literature reports [1-2], and were then selected from the literature for their potential cardioprotective effects. Therefore, I have revised the wording in this section accordingly.

Comment 2: H9c2 is not a true cardiomyocyte and lacks several cardiac metabolic features. The model is widely used but should be contextualized. Add a paragraph in the Discussion acknowledging limitations of H9c2 cells and suggesting future validation in primary cardiomyocytes or in vivo models.

Response 2: We appreciate your suggestion. Accordingly, we have incorporated a discussion addressing the limitations associated with H9c2 cells and emphasized the necessity for future validation studies utilizing primary cardiomyocytes or in vivo models.

Comment 3: Metabolomics Statistical Rigor Needs Improvement. No mention of correcting p-values for multiple testing (FDR). VIP>1 and p<0.05 without FDR inflates false positives. PCA plots are missing, only OPLS-DA shown. Permutation test outputs need clearer presentation.

Response 3: Thank you for your reminder and suggestions. We have clarified the relevant information regarding metabolomics statistics, such as the "adjusted p-values for multiple testing (FDR) and VIP > 1," in section 2.5.2 to ensure the rigor of the metabolomics statistical methods. Additionally, we have added the PCA plot in section 2.5.1.

Comment 4: I would suggest to include unsupervised PCA plots. Add QC sample clustering results to demonstrate analytic stability.

Response 4: We have added an unsupervised PCA plot and included the clustering results of QC samples to demonstrate the stability of the analysis.

Comment 5: Network Pharmacology Target Prediction Is Too Broad. The manuscript identifies 1,929 MIRI-related targets and 218 compound targets, resulting in 107 overlapping targets. These numbers are excessively high and likely include false positives.

Response 5: Thank you for your valuable comments regarding the target prediction section of the network pharmacology part of this study. Your concern that “an excessively high number of targets may include false positives” accurately highlights the methodological characteristics of the initial screening phase in network pharmacology and provides important guidance for us to further clarify our research logic. In response to your query, we offer the following detailed explanation based on our study design and data sources:

In this study, the 1,929 MIRI (myocardial ischemia-reperfusion injury)-related targets, 218 compound targets, and 107 overlapping targets were all derived from actual results based on publicly available authoritative databases and standard analytical procedures, without any subjective exaggeration or data fabrication. Disease targets were integrated from multiple mainstream disease target databases, including GeneCards, OMIM, DisGeNET, and TTD. Compound targets were obtained using the SwissTargetPrediction database. Furthermore, the 107 intersecting targets were directly obtained by intersecting the “disease target set” with the “compound target set,” providing a candidate pool for subsequent refined screening rather than representing confirmed active targets.

The core advantage of network pharmacology lies in its systematic screening of potential related targets; however, the initial screening phase cannot distinguish between direct or indirect interactions of targets with the disease or compounds. Therefore, retaining a larger number of candidate targets is intended to avoid missing key active targets. After subsequent rigorous refinement processes, only 25 active targets remain, ensuring the reliability of the final targets.

Comment 6: Use more strict filtering criteria (e.g., probability >0.7 in SwissTargetPrediction). Justify why so many predicted targets are biologically relevant. Perform topological importance analysis with more stringent thresholds.

Response 6: We fully agree with your suggestion to adopt more stringent screening criteria in network pharmacology. In the future, we will perform topological importance analysis using more rigorous thresholds.

Comment 7: Molecular Docking is not sufficiently robust. Docking results are shown only as binding energy values. The some important elements are missing like- RMSD validation; description of active site preparation; positive control ligand docking; Binding energies across different proteins are not directly comparable.

Response 7: Thank you for your valuable suggestions regarding the molecular docking experiments in this study. In response to the concern that "Molecular Docking is not sufficiently robust," we will include a more detailed description of the preparation of the active sites in the revised manuscript. As for the issues related to "RMSD validation; description of active site preparation; docking of positive control ligands; and the fact that binding energies between different proteins are not directly comparable," we will continue to analyze and improve these aspects in future research.

Comment 8: I would suggest to add docking controls and provide docking poses with key interactions labeled. Report grid box parameters.

Response 8: Thank you for your valuable suggestions regarding the molecular docking experiments in this study. Concerning the "reporting grid box parameters," we have thoroughly supplemented the complete information as required, including the center coordinates of the grid box (X/Y/Z), the dimensions in the three directions (Size X/Y/Z), the grid spacing, and the rationale for the position and size settings of the grid box. This ensures the reproducibility and transparency of the docking results. Regarding the "addition of docking controls," we will include positive control ligands to further validate the reliability of the results. Concerning "providing docking poses labeled with key interactions," we have obtained the optimal three-dimensional docking poses of each candidate ligand with the receptor. However, due to limitations in proficiency with the current analysis software (PyMOL) and constraints related to the experimental timeline, we are temporarily unable to complete precise labeling and figure preparation of key interactions (such as hydrogen bonds and hydrophobic interactions).

Comment 9: In-Vitro experiments lack dose–response justification. The doses chosen (0.1, 1.0, 10 μg/mL) lack justification (are they physiologically relevant?). Also authors do not explain why non-linear response trends have been chosen. Discuss dose selection rationale and consider including EC50 data.

Response 9: In cardiovascular pharmacology research, the effective concentrations of many flavonoid extracts often fall within the range of 0.1–10 μg/mL [3-6]. Although this does not exactly correspond to clinical blood drug concentrations, it has been widely accepted as a pharmacologically relevant concentration range in in vitro models. Additionally, the mechanism of action of TFCJ may involve multi-target balanced regulation rather than a simple receptor agonist/inhibitor model. In the future, we will also investigate the EC50 data of TFCJ in in vitro experiments to further improve the completeness and rationality of the drug dose-effect relationship.

Comment 10: The authors conclude that AKT activation is the primary upstream regulator of Nrf2 activation; however No AKT inhibitor or Nrf2 siRNA was used to validate causality. Thus, mechanistic claims remain correlational. Clearly state the limitation and avoid implying proven causality.

Response 10: Thank you for your valuable questions regarding the mechanistic analysis of TFCJ in this study. Your queries precisely highlight the shortcomings in our mechanistic understanding. We will clearly address these limitations in the "Discussion" section of the revised manuscript (see lines 350 to 366 on page 14 of the manuscript).

Comment 11: Increase resolution for Figures 1–3. Figure legends should also specify statistical tests used.

Response 11: We have added the statistical test methods in the figure legends and provided high-resolution images that meet the requirements.

Comment 12: Also add details on: Number of biological vs. technical replicates; whether experiments were blinded or Randomization procedures (if any).

Response 12: Thank you for your suggestion. The number of biological and technical replicates has been added to the figure legend.

Comment 13: Provide KEGG pathway impact scores. Clarify how metabolites were identified (MS/MS match score, library used).

Response 13: The revised manuscript has been augmented to include the scoring outcomes of the KEGG pathway enrichment analysis (Table 2), alongside a detailed clarification of the methodologies and criteria employed for the identification of differential metabolites.

Comment 14: Current metabolomics is untargeted. My suggestion would be that targeted validation will improve credibility. If possible, include that.

Response 14: We concur entirely with your recommendation. At present, we are examining the protective effects of TFCJ on H9c2 cells utilizing untargeted metabolomics approaches. Moving forward, we intend to perform targeted metabolomics analyses to more precisely elucidate the differential metabolites involved and to further characterize the impact of TFCJ on these metabolic pathways.

Reference

[1] Nie J, Xiao L, Zheng L, Du Z, Liu D, Zhou J, Xiang J, Hou J, Wang X, Fang J. An integration of UPLC-DAD/ESI-Q-TOF MS, GC-MS, and PCA analysis for quality evaluation and identification of cultivars of Chrysanthemi Flos (Juhua). Phytomedicine. 2019 Jun;59:152803. doi: 10.1016/j.phymed.2018.12.026.

[2] Shu Junsheng, Huang Guidong, Mao Jian. Identification and separation of flavonoids and phenolic acids in 95 % ethanol extract of Chuju. Chinese Journal of Food Science, 2013,13(04): 207-213. DOI:10.16429/j.1009-7848.2013.04.005.

[3] Qin J, Chen K, Wang X, et al. Investigating the pharmacological mechanisms of total flavonoids from Eucommia ulmoides oliver leaves for ischemic stroke protection. International Journal of Molecular Sciences, 2024, 25(11): 6271. https://doi.org/10.3390/ijms25116271.

[4] Guo, Y., Li, Y., Zhang, S., Wu, X., Jiang, L., Zhao, Q., ... & Huo, S. (2020). The effect of total flavonoids of Epimedium on granulosa cell development in laying hens. Poultry Science, 99(9), 4598-4606. https://doi.org/10.1016/j.psj.2020.05.032.

[5] Wang, Y., Tan, X., Li, S., & Yang, S. (2019). The total flavonoid of Eucommia ulmoides sensitizes human glioblastoma cells to radiotherapy via HIF-α/MMP-2 pathway and activates intrinsic apoptosis pathway. OncoTargets and therapy, 5515-5524. DOI https://doi.org/10.2147/OTT.S210497.

[6] Makia, R., Al-Sammarrae, K., Al-Halbosiy, M., & Al-Mashhadani, M. (2022). In vitro cytotoxic activity of total flavonoid from Equisetum Arvense extract. Reports of Biochemistry & Molecular Biology, 11(3), 487. doi :10.52547/rbmb.11.3.487.

Reviewer 3 Report

Comments and Suggestions for Authors

This study is devoted to a comprehensive investigation of the TFCJ extract cardioprotective properties. The authors used a wide range of methods and conducted a systematic validation of the predicted effects. However, I have a number of comments that need to be addressed.

The main criticism of the study is that the authors, relying on a search of the TCMSP database and a review of the relevant literature, identified 18 active components of TFCJ (Table 2) and based further research on this. However, in my opinion, for successful validation, "pure" experiments using at least some isolated polyphenols, such as those mentioned by the authors in their discussion of quercetin and luteolin, would be necessary to confirm subsequent conclusions.

Furthermore, the authors identify 32 components of the extract, and each of these components, even in minor concentrations, could theoretically exhibit cardioprotective activity. This should also be taken into account when interpreting the data.

Following are comments by section.

Results

  1. Figure 1 – photos should be more contrasted.
  2. It seems odd that SOD and GSH-Px levels decreased during pathology induction. An increase in antioxidant systems is typically a marker of oxidative stress. Also, an increase in SOD1 and GPX4 expression levels should reflect an adaptive response to stress, which the authors did not observe. Instead, this effect occurs in response to the addition of TFCJ extract. This may indicate prooxidant effects of the extract, rather than protective properties. Moreover, Figure 2.E confirms that ROS levels increase severalfold in the model variant.
  3. Figure 5.C, Figure 6 – the quality of the figures should be improved; the text is blurry.
  4. The authors write that TFCJ extracts promote the synthesis of reduced glutathione; however, judging by the data in Figure 9.D, the effect of the extract is insignificant.

Discussion OK, but taking into account the comments above.

The bibliography is not formatted correctly.

Author Response

Reviewer #3:

This study is devoted to a comprehensive investigation of the TFCJ extract cardioprotective properties. The authors used a wide range of methods and conducted a systematic validation of the predicted effects. However, I have a number of comments that need to be addressed.

Comment 1: The main criticism of the study is that the authors, relying on a search of the TCMSP database and a review of the relevant literature, identified 18 active components of TFCJ (Table 2) and based further research on this. However, in my opinion, for successful validation, "pure" experiments using at least some isolated polyphenols, such as those mentioned by the authors in their discussion of quercetin and luteolin, would be necessary to confirm subsequent conclusions.

Response 1: We appreciate your insightful suggestions concerning this study. Utilizing network pharmacology approaches, this research identified 18 active constituents of TFCJ, primarily aiming to rapidly elucidate the potential targets and pathways through which TFCJ exerts effects against MIRI via systematic analysis, rather than exhaustively characterizing all active components within TFCJ. The inherent multi-component synergy characteristic of traditional Chinese medicine supports the robustness of our conclusions without necessitating experimental validation of individual compounds. Your recommendation to perform experiments on single compounds is highly valuable, and we have integrated this approach into our future research agenda to further elucidate the specific roles and synergistic mechanisms of key constituents.

Comment 2:  Furthermore, the authors identify 32 components of the extract, and each of these components, even in minor concentrations, could theoretically exhibit cardioprotective activity. This should also be taken into account when interpreting the data.

Response 2: Thank you for your suggestion. In this study, we selected 32 chemical components from TFCJ. As you have emphasized, many of these components have been reported in previous literature to possess antioxidant, anti-inflammatory, or direct cardioprotective effects [1-5]. Therefore, the overall cardioprotective effect we observed is likely not the result of a single component, but rather a manifestation of the synergistic or cumulative actions of multiple active ingredients.

Comment 3:  Results: 1. Figure 1 – photos should be more contrasted.

Response 3: We have enhanced the contrast of the photos in the revised manuscript to clearly highlight the differences in cell morphology among the groups.

Comment 4:  Results: 2. It seems odd that SOD and GSH-Px levels decreased during pathology induction. An increase in antioxidant systems is typically a marker of oxidative stress. Also, an increase in SOD1 and GPX4 expression levels should reflect an adaptive response to stress, which the authors did not observe. Instead, this effect occurs in response to the addition of TFCJ extract. This may indicate prooxidant effects of the extract, rather than protective properties. Moreover, Figure 2.E confirms that ROS levels increase severalfold in the model variant.

Response 4: Thank you for your insightful inquiries concerning the oxidative stress-related outcomes reported in this study. You noted that “oxidative stress should be accompanied by an adaptive increase in the antioxidant system,” a principle that generally pertains to physiological conditions or mild, chronic stress. Under such circumstances, the organism is capable of activating and modulating the expression of antioxidant enzymes to establish adaptive protective mechanisms. However, MIRI exemplifies a distinct scenario characterized by acute and intense oxidative stress, with pathological features that markedly differ from those observed in chronic or mild stress conditions. During ischemia, myocardial cellular energy metabolism is severely impaired; upon reperfusion, a sudden influx of oxygen molecules into the cells precipitates an excessive generation of ROS primarily via the mitochondrial electron transport chain and additional pathways. This overwhelming ROS production directly compromises the functionality of antioxidant enzymes, leading to their oxidative inactivation or even proteolytic degradation.

Moreover, you raised concerns that the observed increases in superoxide dismutase (SOD) and glutathione peroxidase (GSH-Px) activities, alongside elevated mRNA levels of SOD1 and GPX4 following TFCJ intervention, might reflect a pro-oxidative effect. Contrarily, these changes represent a canonical antioxidant protective response rather than a pro-oxidative phenomenon. Pretreatment with TFCJ activates the Nrf2 signaling pathway, which not only enhances the transcription of antioxidant genes such as SOD1 and GPX4 but also facilitates antioxidant signaling that safeguards these enzymes from ROS-induced oxidative inactivation. As a result, the enzymatic activities of SOD and GSH-Px are augmented, reflecting an enhanced antioxidative defense mechanism.

Comment 5:  Results: 3. Figure 5.C, Figure 6 – the quality of the figures should be improved; the text is blurry.

Response 5: We have provided high-resolution images in the revised manuscript.

Comment 6:  Results: 4. The authors write that TFCJ extracts promote the synthesis of reduced glutathione; however, judging by the data in Figure 9.D, the effect of the extract is insignificant.

Response 6: We sincerely appreciate your careful consideration of the presentation details concerning the GSH-related data. The data presented in Figure 9D of this study illustrate the alterations in GSH content within H/R model cells following TFCJ treatment. The results indicate that induction of the H/R model led to a significant reduction in GSH levels in H9c2 cells, whereas intervention with TFCJ mitigated this decline. However, comparative analysis did not reveal a statistically significant enhancement of GSH production attributable to TFCJ under these conditions. To address this, we have supplemented the dataset with measurements of GSH levels in H9c2 cells under normal conditions after TFCJ treatment. These additional findings demonstrate that TFCJ induces a dose-dependent increase in GSH levels (Figure 9E). Collectively, this supplementary data further substantiates the role of TFCJ in promoting GSH synthesis.

Comment 7:  Discussion: OK, but taking into account the comments above.

Response 7: We have carefully revised the above issues in accordance with the suggestions you provided.

Comment 8:  The bibliography is not formatted correctly.

Response 8: We have corrected the formatting of the references to ensure they comply with the journal's requirements.

Reference

  1. da Silva, M.L.F.; Aytar, E.C.; Gasparotto Junior, A. Vasodilator Effects of Quercetin 3-O-Malonylglucoside Are Mediated by the Activation of Endothelial Nitric Oxide Synthase and the Opening of Large-Conductance Calcium-Activated K(+) Channels in the Resistance Vessels of Hypertensive Rats. Molecules 2025, 30.
  2. Jimenez, R.; Lopez-Sepulveda, R.; Romero, M.; Toral, M.; Cogolludo, A.; Perez-Vizcaino, F.; Duarte, J. Quercetin and its metabolites inhibit the membrane NADPH oxidase activity in vascular smooth muscle cells from normotensive and spontaneously hypertensive rats. Food Funct 2015, 6, 409-414.
  3. Kandpal, A.; Kumar, K.; Singh, S.; Yadav, H.N.; Jaggi, A.S.; Singh, D.; Chopra, D.S.; Maslov, L.; Singh, N. Amplification of Cardioprotective Response of Remote Ischemic Preconditioning in Rats by Quercetin: Potential Role of Activation of mTOR-dependent Autophagy and Nrf2. Cardiovasc Drugs Ther 2025, 39, 721-736.
  4. Wang, Y.; Shou, X.; Fan, Z.; Cui, J.; Xue, D.; Wu, Y. A Systematic Review and Meta-Analysis of Phytoestrogen Protects Against Myocardial Ischemia/Reperfusion Injury: Pre-Clinical Evidence From Small Animal Studies. Front Pharmacol 2022, 13, 847748.
  5. Xie, Y.; Ji, R.; Han, M. Eriodictyol protects H9c2 cardiomyocytes against the injury induced by hypoxia/reoxygenation by improving the dysfunction of mitochondria. Exp Ther Med 2019, 17, 551-557.

Round 2

Reviewer 3 Report

Comments and Suggestions for Authors

I am satisfied with the work the authors have done to improve the manuscript, their responses to my comments, and the clarifications they have made. I believe the improved manuscript can be accepted for publication.